# Chia Oil Microencapsulation Using Tannic Acid and Soy Protein Isolate as Wall Materials

**DOI:** 10.3390/foods12203833

**Published:** 2023-10-19

**Authors:** Paola Alejandra Gimenez, Agustín Lucini Mas, Pablo Daniel Ribotta, Marcela Lilian Martínez, Agustín González

**Affiliations:** 1Departamento de Química Orgánica, Facultad de Ciencias Químicas, Universidad Nacional de Córdoba, Córdoba X5000GYA, Argentina; paolagimenez17@mi.unc.edu.ar; 2CONICET, Instituto de Investigación y Desarrollo en Ingeniería de Procesos y Química Aplicada (IPQA), Córdoba X5016GCA, Argentina; 3CONICET, Instituto de Ciencia y Tecnología de Alimentos Córdoba (ICYTAC), Córdoba X5016GCA, Argentina; agustin.lucini@unc.edu.ar (A.L.M.); pdribotta@unc.edu.ar (P.D.R.); 4Facultad de Ciencias Exactas, Físicas y Naturales, Universidad Nacional de Córdoba, Córdoba X5000GYA, Argentina; 5CONICET, Instituto Multidisciplinario de Biología Vegetal (IMBIV), Córdoba U9120ACD, Argentina

**Keywords:** antioxidants, chia oil, cross-linking, microencapsulation, polyphenols, fatty acids

## Abstract

The use of proteins to produce oil-containing microcapsules has been previously analyzed; however, their chemical modification, in order to improve their performance as wall materials, is a strategy that has not been widely developed yet. This study aimed to analyze the chemical modification of the proteins through cross-linking reactions with tannic acid and to evaluate their performance as wall materials to the microencapsulation of oils rich in polyunsaturated fatty acids. The cross-linking reaction of isolated soy protein and tannic acid was carried out at pH 10–11 and 60 °C. Subsequently, emulsions were made with a high-speed homogenizer and microcapsules were obtained by spray drying. Microcapsules were characterized by particle size, morphology (SEM), total pore area and % porosity (mercury intrusion methodology), superficial properties (contact angle), and size distribution of oil droplets (by laser diffraction). Additionally, encapsulation efficiency was determined as a function of total and surface oil. Oil chemical stability and quality were studied by Rancimat, hydroperoxide values, and fatty acid profiles. In addition, a storage test was performed for 180 days, and released oil and polyphenols were determined by in vitro gastric digestion. Moreover, the fatty acid composition of the oil and the total polyphenol content and antioxidant capacity of polyphenols were analyzed. The results showed that spray-dried microcapsules had an encapsulation efficiency between 54 and 78%. The oxidative stability exhibited a positive correlation between the amount of polyphenols used and the induction time, with a maximum of 27 h. The storage assay showed that the peroxide value was lower for those cross-linked microcapsules concerning control after 180 days. After the storage time, the omega-3 content was reduced by 49% for soy protein samples, while cross-linked microcapsules maintained the initial concentration. The in-vitro digestion assay showed a decrease in the amount of oil released from the cross-linked microcapsules and an increase in the amount of polyphenols and a higher antioxidant capacity for all samples (for example, 238.10 mgGAE/g and 554.22 mg TE/g for undigested microcapsules with TA 40% versus 322.09 mgGAE/g and 663.61 mg TE/g for digested samples). The microcapsules showed a high degree of protection of the encapsulated oil, providing a high content of polyunsaturated fatty acids (PUFAS) and polyphenols even in prolonged storage times.

## 1. Introduction

Chia seed oil contains around 61–70% alpha-linolenic acid (18:3), which makes it the largest source of omega-3 fatty acid in vegetables [1]. The consumption of polyunsaturated fatty acids of the omega-3 series gives several health benefits and can be incorporated as triglycerides or ethyl esters [2]. A diet rich in PUFAS reduces the risk of contracting coronary and neurodegenerative diseases, cancer, metabolic syndrome, rheumatism, type 2 diabetes, atherosclerosis, and Alzheimer’s disease [2]. Even though the consumption of omega-3 fatty acids presents nutritional benefits, some disadvantages are present due to their poor oxidative stability and shortened shelf life. One of the main challenges for the use and incorporation of these PUFAS-rich oils in processed foods is the need for them to be stabilized by incorporating antioxidant compounds or transported in a polymeric matrix that contains and protects them. A suitable technology for this goal is microencapsulation in natural polymeric matrices [3]. Several methodologies can be founded in the literature such as freeze-drying, coacervation, ionic gelation, and impregnation; however, spray-drying is the most explored methodology applied to food and cosmetic industry due to its versatility for encapsulating labile compounds [4].

The use of plant proteins as wall material is being widely developed. Microencapsulation with soy protein isolate (SPI) and protein/polysaccharide combinations has been studied, showing a protective result generated from the wall material which preserves the quality of the oil at least for 90 days [3]; however, it is necessary to study strategies to obtain microcapsules that can exert a prolonged protective effect on the bioactive compound to make it viable on an industrial scale compared to those reported until now [5]. Cross-linking treatment seems to be a novel and appropriate strategy for obtaining a more rigid and less permeable wall material.

The cross-linker agents most extensively used for proteins are aldehydic compounds such as glutaraldehyde, formaldehyde, glyoxal, and phenolic compounds. Nevertheless, the cytotoxicity of these compounds restricts their use for food applications [6].

Tannic acid (TA), is a low-cost polyphenol of plant origin with OH groups and aromatic rings [7] which are highly reactive to proteins and amino acids and have functional properties such as antioxidant, and antimicrobial properties and prevent various chronic diseases [8,9]. The phenolic compounds can be oxidized to their quinone counterparts in alkaline medium, which are capable of reacting with the free amino groups of protein, resulting in products with enhanced antioxidant activity with the ability to bind proteins by hydrophobic/hydrogen interactions or covalent C-N bonds under alkaline conditions and the presence of molecular oxygen [7,10].

In addition, it was reported an increase of about 67% in the antioxidant power of tannic acid after 15 min of thermal treatment. This effect was reported to be associated with the free gallic acid and a galloyl group in the remaining gallotannins produced after thermal hydrolysis [11,12].

Based on this, we hypothesize that microcapsules prepared with a cross-linked wall material could have a more closed structure and fewer pores through which the oil can diffuse to the outside, increasing the protective effect. In addition, the inherent antioxidant capacity of the cross-linker polyphenol can exert an additional protection effect. All these factors would help to increase the protection provided on the oil by this cross-linked microcapsule compared to non-cross-linked materials.

Therefore, our work aimed to study an interesting and novel strategy such as the chemical modification of the soy proteins used as wall material through cross-linking reactions with TA.

As was explained, several previous studies focus on the microencapsulation process, dehydration methods, use of antioxidants, or other aspects, however, this work presents a novel study based on the chemical modification of the wall material and its influence on the oil stability. The development of new cross-linked wall materials for the microencapsulation of oils rich in PUFAS could protect and extend the shelf life of these active components, improving its performance in foods whose production process requires working conditions that affect their stability and enabling the production of new omega-3-enriched foods. This development would produce a great impact on the food industry. The consumption of these foods would contribute to maintaining a healthier diet while maintaining more appropriate omega-3 levels, which would help to prevent certain cardiovascular diseases.

## 2. Materials and Methods

### 2.1. Materials

Soy protein isolate (SPI) (90% protein, fat-free dry basis) was purchased from DuPont Nutrition & Health (USA). Chia seeds were obtained from the province of Salta, Argentina (Nicco S.R.L., Córdoba, Argentina). The reagents and solvents used were analytical grade and were purchased from local distributors.

### 2.2. Oil Extraction and Preparation of Microcapsules

Five hundred milliliters of CO was extracted by cold-pressing in a single step with a Komet screw-press (CA 59 G, IBG Monforts, Mönchengladbach, Germany) according to Martinez et al. [13].

Aqueous dispersions of SPI 8% *w*/*w* were made and brought to pH 10–11. Both TA and heat-treated TA (130 °C in the autoclave for 15 and 30 min [12] (TA15 and TA30) were studied. Fifteen samples (and their triplicates) were prepared with different amounts of TA, TA15, and TA30 (1–40% *w*/*w*, with respect to SPI content) and allowed to react by shaking at 60 °C. Aliquots were taken for different periods of time (1, 2, 3, 6, and 24 h). [7]. CO was incorporated dropwise into the dispersions at a 2:1 ratio (SPI: oil) for 15 min at 18.000 rpm using an Ultraturrax homogenizer (IKA T18, Burladingen, Germany). The emulsions were dried in a Mini Spray Dryer Büchi B-290 (Büchi Labortechnik, Flawil, Switzerland) under the following optimized conditions: inlet air temperature: 130 °C; air outlet temperature: 80 ± 1 °C; air atomizing pressure: 4.4 L/h; atomization air flow: 538 L/h; pump regulation: 10%; and suction regulation: 100% [3].

#### Quantification of Cross-Linker Consumption

The amount of TA, TA15, and TA30 consumed in the cross-linking reaction was determined by UV/vis spectrophotometry (Shimadzu UV-1800, Kyoto, Japan) quantifying the unreacted cross-linker [7] that was removed from the dispersions by dialysis for 2 days using dialysis bags with a molecular weight cut-off of 14 kDa.

### 2.3. Characterization of Microcapsules

#### 2.3.1. Morphological Analysis

Scanning electron microscopy (SEM) (FEG-SEM, Carl Zeiss-Sigma, Roedermark, Germany) and confocal fluorescence microscopy (LSM5 Pascal; Zeiss, Aalen, Germany) were used to study the particle size and morphology of the microcapsules following a previously reported methodology [3]. Oil distribution was visualized by staining samples with Nile Red (1 g/kg fat basis). The excitation was done using a 488 nm argon-ion laser and detection through a 515/30 filter [14].

The porosimetry was determined in terms of total pore area (TPA) and % porosity by mercury intrusion methodology using an Autopore III 9410 apparatus (Micromeritics, Norcross, GA, USA). The pore area (PA) represents the specific surface of the material (in m^2^/g). The percentage of porosity refers to the number of pores in the material in percentage terms [%] [15].

#### 2.3.2. Contact Angle

The contact angle was measured using deionized water in a contact angle goniometer following the methodology proposed by Romero et al. [16]. The contact between the droplet of deionized water and the surface formed by compressed microcapsules was recorded on video using a CMOS camera at 15 frames per second. Each video was processed and different frames were selected. Finally, the images were converted to 8-bit grayscale using IMAGEJ 1.4 g software. The contact angle was measured by processing the images with the LB-ABSA (Aurélien Stalder Low-Binding Aximetric Droplet Shape Analysis) plugin. The results obtained correspond to the contact angle by the subpixel method. All assays were performed at room temperature (25 °C).

#### 2.3.3. Size Distribution of Oil Droplets in Reconstituted Emulsions

The size distribution of oil droplets was determined by laser diffraction according to Bordon et al. [17] with a LA 950V2 Horiba analyzer (Kyoto, Japan). The relative refractive index used (refractive index of oil, 1.47/refractive index of water, 1.33) was 1.10. *D*[4,3] (De Broucker) mean diameter and the polydispersity index (*PDI*) were determined in triplicate.
(1)$$D\left[4,3\right]=n_{i}\;.\;d_{i}^{4}/n_{i}^{3}{}_{\;},$$
where $n_{i}$ corresponds to the number of droplets of diameter $d_{i}$.
(2)$$PDI=\left(DV_{90}-D_{10}\right)/\;DV_{50},$$
where 90, 10 and 50% of the oil volume is contained in droplets of diameter below or equal to $DV_{90}$, $D_{10},$ and $\;DV_{50}$, respectively.

#### 2.3.4. Color Measurements

Color measurements were obtained using a colorimeter (Konica-Minolta CM600d, Tokyo, Japan), working with D65 (daylight) and an inclination of 10°. The color parameters were expressed as *L** (lightness), *a** (redness/greenness), and *b** (yellowness/blueness) values [3]. Whiteness (*WI*) and yellowness (*YI*) indexes were calculated [17].
(3)$$WI=L^{*}-3b^{*},$$
(4)$$YI=\left(142.86\;b^{*}\right)/L^{*}.$$

#### 2.3.5. Moisture Content and Water Activity

A moisture analyzer with halogen heating (HE53 Mettler Toledo, Panorama City, CA, USA) was used for the determination of moisture content (MC). Water activity (a_w_) was measured with Aqua-Lab (Decagon Devices Inc., EUA, Pullman, WA, USA) at 25.0 ± 0.5 °C.

#### 2.3.6. Determination of Encapsulation Efficiency

A previously reported methodology [3] was used for the determination of surface or free oil (SO), total oil (TO), and encapsulation efficiency (EE). A Soxhlet extraction for 24 h with 200 mL of *n*-hexane was performed on 4.00 ± 0.10 g of sample. The TO extracted was weighed and expressed as a percentage of oil respect to the weight (d.b.) of the microcapsules. For SO, 2.00 ± 0.01 g of the sample was weighed and mixed with 30 mL of petroleum ether, stirred for 1 min, and filtered. The resulting solids were washed with 10 mL of petroleum ether and the organic phases were combined. The solvent was evaporated and the remaining oil was heated at 105 °C in an oven to constant weight. Encapsulation efficiency (EE) was determined by Equation (5).
(5)EE=(TO−SO)×100/TO


### 2.4. Oil Oxidative Stability Study

The oxidative stability of unencapsulated and encapsulated oil samples were subjected to accelerated oxidation conditions in a Rancimat (METROHM, Herisau, Switzerland) apparatus (100 °C, airflow 20 L/h) and expressed as induction period (IP, h) [3]. The protection factor (PF) was defined as the ratio of the IP of the microencapsulated oil and the IP of the unencapsulated oil.

The hydroperoxide values (HPV) were determined by iodometric titration following an AOCS methodology [18] with some modifications. Briefly, 0.20 ± 0.01 g of the extracted oil was weighed, and 3 mL of acetic acid: chloroform (3:2% *v*/*v*) was added and stirred vigorously until complete dissolution was achieved. Afterwards, 0.5 mL of saturated potassium iodide solution was added and the system was kept in the dark for 1 min. The reaction was stopped by the addition of 3 mL of distilled water, and 0.5 mL of starch solution (1%, *w*/*v*) was added as an indicator. Finally, solutions were titrated with 0.001 N Na_2_SO_3_ until the brown color disappeared. The calculation of HPV was carried out using Equation (6) which is expressed in milliequivalents of oxygen/kg oil.
(6)HPV=(S−B)×N×1000/w
where *S* represents the volume in mL of the sodium thiosulfate solution consumed by the sample, *B* is the volume consumed by the blank, *N* is the normality of sodium thiosulfate solution, and *w* represents the mass of oil expressed in grams.

### 2.5. Fatty Acid Composition

The fatty acid composition of the bulk oil and the oil extracted from the microcapsules (solvent extraction) was analyzed by gas chromatography according to González et al. [14] using C21:0 as an internal standard for the fatty acid quantification.

### 2.6. Storage Test

Unencapsulated CO and microencapsulated oil were placed in a thermostatted chamber at 25 °C in 250 mL amber glass bottles. The samples were stored for 180 days. At different times, 6 g of samples were taken and CO was extracted by immersing them in hexane for 24 h at 4 °C and evaporating the solvent in a vacuum at 36 °C to evaluate their HPV, IP, and fatty acid composition (Section 2.5). Moisture content, water activity, and particle size distribution were determined at the beginning and end of the storage assay.

### 2.7. In Vitro Gastric-Intestinal Digestion

In vitro digestion was simulated according to the proposed by Gañan et al. [19] with modifications. Briefly, 2.00 ± 0.01 g of microcapsules and bulk oil were taken and contacted for 5 min with 4 mL of freshly collected human saliva at 37 °C [20]. The pH was adjusted to 2 with 1 M HCl to stop the action of the amylase. Subsequently, 12.5 mL of simulated gastric fluid (SGF) was added and incubated at 37 °C for 2 h with constant shaking at 60 osc/min to simulate gastric digestion. SGF was prepared with 100 mg of NaCl, 0.35 mL of HCl 36% *w*/*v*, and 160 mg of pepsin in 50 mL of water.

The pH was adjusted to 7 with NaOH 5 M. Next, 12.5 mL of simulated intestinal fluid (SIF) was added and shaken at 40 osc/min for 2 h at 37 °C. SIF was prepared with 340 mg of K_2_HPO_4_, 440 mg of NaCl, and 320 mg of pancreatin in 50 mL of water,

To determine the amount of oil released after digestion, the oil was extracted with three portions of 25 mL of *n*-hexane. The solvent was completely evaporated, weighed (wf), and compared with the amount of initial oil contained in the microcapsules (wi), which was previously determined using the procedure described. The amount of oil available after digestion (%OA) was calculated as Equation (7):


(7)
%AO=wf/wi×100


The fatty acid composition of the oil extracted from the microcapsules was evaluated according to Section 2.5.

#### 2.7.1. Total Polyphenol Content

Total polyphenol content of microcapsules (TPC) was determined using by Folin–Ciocalteu method [21] for digested (from the aqueous phases obtained after the extractions carried out for the digestion test) and undigested samples (microcapsules) in order to investigate if polyphenols are released after the gastrointestinal process. Shortly, the absorbance of appropriately diluted samples with the addition of Folin–Ciocalteu commercial reagent and an aqueous solution of sodium carbonate 20% was measured at 750 nm. TPC was calculated using a calibration curve constructed with gallic acid (GA). Results were expressed as milligrams of polyphenols equivalent to gallic acid per g of sample (mg GAE/g). Blank samples (containing only the reagents) were used to discount the absorbance due to solvents and reagents.

#### 2.7.2. In Vitro Antioxidant Capacity

To determine the antioxidant capacity (AC) of digested (from the aqueous phases obtained after the extractions carried out for the digestion test) and undigested samples (microcapsules), the reducing power and the radical scavenging activity were measured. In vitro antioxidant capacity was determined for undigested and digested samples (from the aqueous phases obtained after the extractions).

##### Reducing Power

The reducing power was measured by the ferric-reducing antioxidant power (FRAP) method [22]. In brief, the adequately diluted sample was mixed with the corresponding reagent prepared with FeCl_3_ and TPTZ in buffer acetate pH = 3.6 and measured at 593 nm. Results were obtained from a calibration curve made using Trolox and expressed as mg of Trolox equivalents per g of sample (mg TE/g). Blank samples (containing all reagents) were used to discount the absorbance due to solvents and reagents.

##### Radical Scavenging Activity

The radical scavenging activity was measured by the methods ABTS [23] and DPPH [24].

First, the properly diluted samples were mixed with the pre-formed radical ABTS•+ (generated by oxidation of ABTS with K_2_S_2_O_8_) and measured the absorbance at 734 nm. On the other hand, samples were also mixed with DPPH• and measured at 515 nm. In both cases, results were obtained from a calibration curve made using Trolox and expressed as mg of Trolox equivalents per g of sample (mg TE/g). Blank samples (containing all reagents) were used to discount the absorbance due to solvents and reagents.

### 2.8. Statistical Analysis

Analytical determinations were the averages of at least triplicate measurements of individual samples prepared in different batches. ANOVA test at the 5% level (*p* < 0.05) of significance was used for the determination of the statistical differences among treatments for all parameters evaluated. The INFOSTAT/Professional 2014 software (FCA-UNC, Córdoba, Argentina) was used.

## 3. Results and Discussion

### 3.1. Quantification of Cross-Linker Consumption

The optimal reaction time was determined as the one in which the greatest amount of TA reacts with the protein. It was measured that with 24 h of reaction, the highest percentage of cross-linking was achieved (56% of bounded TA concerning initially added). Although some authors report reaction times of 1, 2, or 9 h [7,8,10], for this system, low reaction yields were obtained at these shorter times.

A positive correlation was observed between the amount of cross-linker that reacted and the initial quantity added (Figure 1). As expected, very low consumption of cross-linker was observed when they were added in small quantities, but a marked increase was observed when they were added in larger quantities, with the value of 40%presenting the largest cross-linker consumption. It is expected that the first interactions formed between cross-linker and proteins are of the hydrogen bond type. Once these initial interactions are formed, some covalent interactions are formed through the formation of quinones produced by the basic medium. As is known, hydrogen bond interactions are weaker than covalent ones, allowing the removal of these polyphenols from the protein material; however, covalent interactions cannot be broken, with these remaining polyphenols retained in the protein matrix. Perhaps the 40% concentration is high enough to occupy a large number of sites available for hydrogen bonding interactions and to form a high amount of covalent bonds with respect to the other concentrations, which produces a very marked increase in the quantity of cross-linker consumed. In addition, it can be observed that the relative amount of polyphenol consumed for TA, TA15, and TA30 samples is similar for all added concentrations except for 40% where a higher consumption is observed for polyphenols with longer heat treatment (TA30 ˃ TA 15 ˃ TA). This may be due to the fact that the heat treatment produces hydrolysis of the TA producing smaller molecules which can move more freely, which allows them to find the reaction sites more easily. These results were lower than those obtained for matrices with casein [7].

### 3.2. Characterization of Microcapsules

Color characterization of the microcapsules is important considering their potential food application. A characteristic dark green color provided by the reaction of TA with proteins was observed for the powders obtained by spray-drying of the prepared emulsions. The *L** and *WI* values decreased, while the *YI* values increased with the amount of cross-linker initially added (Table 1). The variation in these calculated values indicates an increase in the dark green coloration for the samples with the highest amount of polyphenols, which is in accordance with what was determined by the quantification of the amount of polyphenols consumed since there is a greater proportion of the cross-linking reaction. Similar results were reported by Guo et al. (2021) [10] and Picchio et al. (2018) [7] for soy protein gels and casein films with TA. These authors observed a significant decrease in *L**, *a**, and *b** values with increasing TA content.

Microcapsules moisture content (MC) is shown in Table 2. MC plays a significant role in establishing the shelf-life of microcapsules. It is expected that the samples that have a greater amount of polyphenols present a higher moisture content due to the polar characteristics of the TA; however, marked variations or a clear trend of the humidity in the different samples could not be observed. The moisture content of 3–4% is the minimum specification for most dried powders used in the food industry [25]. Water activity (a_w_) was between 0.16 and 0.18; under these conditions, lipid oxidation and microbial proliferation are reduced [26]. These results were similar to those reported in the literature [3,27,28]. Regarding the distribution of oil in the microcapsules, the encapsulation efficiency (EE) was between 54 and 78% (Table 2). Important deviations are observed, which does not make it possible to analyze variations between samples. Similar results were obtained by González et al. (2016) [3], Timilsena et al. (2016) [29], and Bordon et al. (2022) [27] for SPI and maltodextrin, chia seed protein isolate and chia seed gum, and SPI and gum arabic microcapsules with values among 52–65%, 67–93%, and 68–87%, respectively.

To analyze if the cross-linking can affect the porosity of the wall material, a mercury intrusion porosimetry methodology was performed for samples that contain the maximum amount of cross-linker with respect to samples without cross-linker, and results can be seen in Table 3. It could be observed that no statistically significant differences among samples were obtained; nevertheless, a decreasing trend of porosity percentage and total pore area (TPA) in the samples with 40% of polyphenols was identified. This result can be attributed to a more compact matrix acquired due to a major percentage of cross-linkers in the wall material used.

The microstructure of microcapsules was studied by different microscopies. The SEM micrographs and confocal microscopy images are shown in Figure 2. The microcapsules presented a spherical morphology with depressions, without fractures or macropores, and a particle size between 2 and 8 μm. No marked differences could be observed among samples. From the micrographs could be observed that microcapsules can occur individually or may form aggregates. Fluorescence confocal microscopy using Nile Red to stain the lipid phase showed that the oil droplets are homogeneously distributed inside the microcapsule (Figure 2). These results were similar to those reported by Gonzalez et al. (2016) [3] but contrary to what was observed by (Di Giorgio et al., 2019) [28], where soybean proteins were co-localized with the fish oil drops.

The oil droplet size distribution in reconstituted emulsions was analyzed. Mean diameter (*D*[4,3]) and polydispersity index (PDI) values are shown in Table 2. These results were similar to those reported for SPI and gum arabic [27]. Some samples exhibited a unimodal distribution (samples with PDI < 1), while others exhibited a bimodal distribution (samples with PDI > 1). No clear trends were observed regarding the oil droplet size distribution, which may be attributed to the fact that the emulsions were prepared using only a high-speed homogenizer [17].

### 3.3. Oxidative Stability of Microcapsules

The protective effect of microcapsules with cross-linked wall materials on the oxidative stability of CO was demonstrated by obtaining long induction periods in the Rancimat test (Table 2). Microencapsulated oil only with SPI without polyphenols in the wall material showed a larger IP with respect to free oil, demonstrating the protective effect of the microencapsulation process. In addition, when CO was microencapsulated using polyphenols as wall material cross-linkers, the IP significantly increased (*p* < 0.05). Indeed, the higher the proportion of cross-linking polyphenol used, the higher the IP obtained. The maximum induction period (IP) and protection factor (PF) was around 27 h and 11 for 40% of cross-linker agents, respectively (Table 2). These results were higher than those reported by González et al. (2016) [3] with a maximum of 6.4 h and 2.7, respectively. The oxidative stability of the microencapsulated CO with SPI (without cross-linking) in the present study was lower compared with González et al. [3]. This difference may be due to the fact that in that research the authors perform a heating step in the preparation of the wall material, which produces compounds with antioxidant activity as a result of the Maillard reaction [30]. For spray-dried microcapsules obtained by complex coacervation of chia seed protein isolate and chia seed gum, IP ~12 h was obtained [29].

Furthermore, the antioxidant power of the natural polyphenols was assessed by adding the same amount of TA in bulk CO concerning encapsulated oils. The IP and PF obtained for bulk CO with 5% and 40% of polyphenols were 5.6 h and 11.0 h, and 2.3 and 4.4, respectively. These results demonstrated that polyphenols exert an antioxidant effect, but it is smaller than the protection obtained when polyphenols are part of the microcapsule wall material. It could be observed that the simple addition of both effects separately as microencapsulation (sample with microencapsulated oil in SPI wall material without TA) and the antioxidant effect (bulk CO with TA samples) are smaller than the combination of both strategies (microencapsulated oil in cross-linked SPI wall material). From this, it can be concluded that there is a synergistic effect between both protection strategies.

### 3.4. Storage Test

For this assay, microcapsules with 5 and 40% cross-linker agents TA and TA15 were selected. These samples were chosen to compare treatments with maximum and medium protection effects. Figure 3 shows that hydroperoxide values (HPV) gradually increased with storage time. After 180 days, samples with 40% polyphenols presented lower HPV values than the maximum limits established in the Codex Alimentarius for cold-pressed and virgin vegetable oils (15 meq. of O_2_/kg oil) [31]. On the other hand, samples with 5% polyphenols presented higher HPV values than the maximum established by the legislation, but these values were lower than the once observed for the microencapsulated samples without wall material chemical modification (SPI sample). Other studies have reported higher oxidation rates after 30 days for CO microencapsulated with chia seed protein isolate and chia seed gum [29] and after 90 days with SPI [3] and SPI/gum arabic as wall materials [27].

In addition, Figure 3 shows that the induction period decreased with storage time for all the samples evaluated; however, MC with 40% cross-linking agent presented longer induction periods (11.3 and 13.3 h for AT and AT15, respectively) than MC with 5% or no cross-linking agent. These results indicate that after 180 days of storage, the microcapsules formulated with the highest polyphenol proportion exert a considerable protective effect on CO oxidative stability. Bordon et al. (2022) [27] and Copado et al. (2021) [26] informed a considerable reduction in the IP after 90 and 30 days of storage for CO microcapsules formulated with SPI and gum arabic and hydrolyzed sunflower lecithins, chitosan, and chia mucilage, respectively.

The results of the fatty acid profile on day 0 were in compliance with those reported in the literature [32] for cold-pressed chia oil, observing that the microencapsulation process did not affect CO fatty acid composition (Table 4). In addition, the storage assay demonstrated that the microcapsules with the cross-linked wall material (TA 40% and TA15 40%) were able to maintain omega-3 content, while the non-cross-linked samples decreased their content to 49% after 180 days. The results obtained for the cross-linked samples were in concordance with those reported by Copado et al. (2021) [26] for multilayer microcapsules. Furthermore, an increase in aw was observed during the storage test, with the SPI microcapsules being the ones that showed the greatest increase. The increment in aw has been reported [26]. The moisture was higher for all the samples without a clear trend. In addition, the particle size distribution showed a large increase for the SPI samples after the storage time, while for the microcapsules with cross-linked wall material, a slight increase was measured and no differences were observed between samples with 5 or 40%. In this sense, Ixtaina et al. (2015) [33] reported an increase in particle size after storage for sodium caseinate-lactose-based microcapsules containing chia oil. This increase in particle size corresponded to the presence of conglomerates caused by an increase in surface oil. This was corroborated by the results obtained by contact angle determinations. These results showed that the SPI samples presented initially a hydrophilic surface (83.13°), and after storage time they became hydrophobic (134.52°), whereas the microcapsules with cross-linked wall material remained hydrophilic (angle < 90°) after 180 days (74.52 ± 3.72 and 79.23 ± 4.62 for TA 40% and TA15 40%, respectively) [34]. The result obtained for SPI (day 0) was in concordance with Locali Pereira et al. (2019) [34], who reported contact angles between 84 and 89° for spray-dried microcapsules of pink pepper using SPI, maltodextrin, and high methoxyl pectin as wall material. The contact angle measurements demonstrated that a minor oil diffusion from the core to the surface of microcapsules occurs for cross-linked wall materials concerning non-cross-linked probably caused for a denser structure derived from the cross-linking process.

### 3.5. In Vitro Gastrointestinal Digestion

In vitro digestion of microcapsules was conducted to evaluate the amount of oil released into the gastrointestinal tract (%OA). Samples that maintained their protective capacity and a hydroperoxide value (HPV) lower than the limit established in the Codex Alimentarius (Codex Alimentarius, 2001) after storage for 180 days, were selected (TA 40% and TA15 40% and SPI and bulk oil as control samples). The results of % AO for microcapsules without cross-linking agents (SPI) showed no significant differences compared to the literature for microcapsules of SPI [14,17,19]. However, microcapsules with TA had significantly lower % AO (*p* < 0.05). Similar results were reported for flaxseed oil microencapsulated with whey protein and sodium caseinate, with oil release percent of 23.12 ± 7.06 and 40.54 ± 1.75% respectively [25].

In the gastrointestinal tract, emulsions were exposed to the surface-active components which help in emulsification [35]. An increase in particle size would reduce the total surface area available for interaction with digestive enzymes [36]. Polyphenols affected emulsions by increasing droplet size and decreasing specific surface area thereby potentially reducing lipase activity and fat absorption [35,37].

This decrease in the amount of released oil is likely due to the complexity of the wall material of the microcapsules. Cross-linked proteins are less soluble concerning non-cross-linked samples, resulting in more difficulty to disrupt the structure of the wall and therefore reaching a slower diffusion of the oil to the medium. In addition, Li et al. (2020) [37] reported that pea protein and TA coating inhibited the ability of bile salts and lipase enzymes to reach the oil-water interface, thus restricting lipid digestion.

On the other hand, the composition of fatty acids for all the digested samples did not show significant differences (*p* > 0.05) between them. In addition, no significant differences were found in fatty acid composition for microencapsulated oils before and after the in-vitro digestion (Table 5). This was in agreement with Gonzalez et al. (2018) [14] and Bordon et al. (2021) [17]. This suggests that the simulated gastrointestinal conditions did not modify the chemical quality of the microencapsulated oils.

Polyphenol content (TPC) and antioxidant capacity (AC) were determined for the undigested and digested samples in order to investigate if polyphenols are released after the gastrointestinal process. These results are summarized in Table 6. As it was expected, SPI showed low values of TPC, which were increased after the gastrointestinal digestion process. These results could be explained since some proteins, peptides, and free amino acids could react with the Folin–Ciocalteu reagent [38]. However, this sample did not present reducing power and radical scavenging activity. On the other hand, the microcapsules with TA and TA15 showed close TPC values to the estimated according to how they were prepared. Nevertheless, after gastrointestinal digestion, all samples exhibited an increment in TPC. Tannic acid is a combination of glucogallic acid with two ester-like bound molecules of gallic acid, so the changes in pH and the action of the enzymes in the gastrointestinal digestion model could liberate different units of gallic acid [39], increasing TPC. Similar trends were observed in FRAP, TEAC, and DPPH measurements. An increase in the reducing power and radical scavenging activity of all the samples analyzed was observed after the digestion process. Badhani et al. (2015) [40] reported that gallic acid has a significantly higher reducing power and antiradical potential than other natural antioxidants such as caffeic acid, ascorbic acid, sinapinic acid, tannic acid, isoeugenol, and sesamol, among others, and, in turn, its antioxidant capacity is affected by its concentration [40,41]. Moreover, the thermal treatment carried out in TA15 treatment helped to increase even more TPC and AC. The increase in antioxidant activity after heat treatment was reported in the literature [11,12]

## 4. Conclusions

The microencapsulation of chia oil using wall materials cross-linked with tannic acid allowed us to produce omega-3-rich powders with a high content of omega-3 fatty acids and bioavailable polyphenols with the ability to maintain the chemical quality of the chia oil for long periods with minimal levels of hydroperoxides, allowing its potential application as an ingredient in processed foods. The microcapsules were obtained by spray drying, and different concentrations of tannic acid with and without thermal treatment were evaluated to identify the matrix with the best protective effect. The induction times of the microencapsulated chia oil were between 3 and 11 times greater than those of the unencapsulated oil under accelerated oxidative conditions. The oxidative stability observed during the storage of chia oil for 180 days demonstrated that the microcapsules exert a protective effect over the oil, which is corroborated with HPV values below the Codex Alimentarius limit (15 meq O_2_/kg of oil) and fatty acids profile derived from obtaining an optimized wall material and from the intrinsic antioxidant properties of these cross-linkers. Even though the in vitro digestion of the microcapsules with TA and TA15 material showed a lower amount of available oil due to wall material modifications, a larger quantity of polyphenols, and a higher reducing power and radical scavenging activity in the digestate were obtained. Therefore, these microcapsules provide the dual benefit of their high content of omega-3 fatty acids and polyphenols with their innumerable already known biological activities.

## Figures and Tables

**Figure 1 foods-12-03833-f001:**
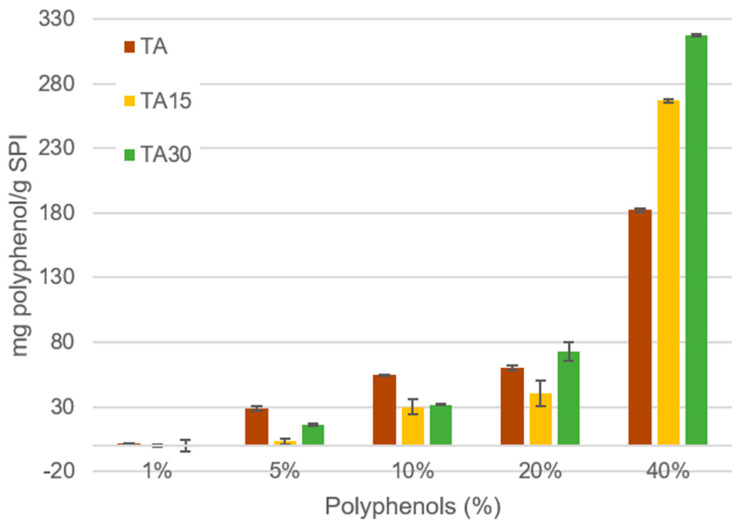
Amount of polyphenols consumed per gram of SPI for the different polyphenols (TA, TA15, and TA30) initially added at different concentrations (1, 5, 10, 20, and 40%).

**Figure 2 foods-12-03833-f002:**
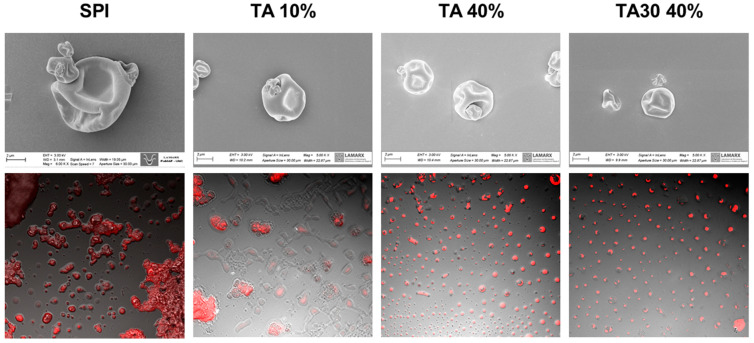
Scanning electron microscopy (SEM) (up) and confocal microscopy micrographs (down) of samples with 0, 10, and 40% TA and thermally treated TA. Figure 2 shows only 4 samples because no differences were found among all samples.

**Figure 3 foods-12-03833-f003:**
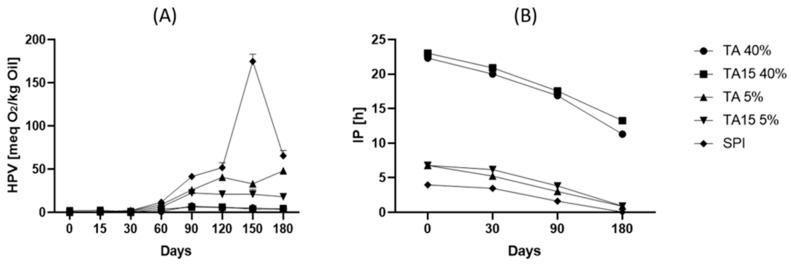
(**A**) Hydroperoxide values (HPV) and (**B**) induction period (IP) at different storage times.

**Table 1 foods-12-03833-t001:** Color parameters of the different samples using CIELab scale.

Samples	*L**	*a**	*b**	*WI*	*YI*
SPI	89.46 ± 0.04 ^L^	−0.05 ± 0.01 ^A^	10.97 ± 0.04 ^A^	56.54 ± 0.09 ^H^	17.52 ± 0.06 ^A^
TA 1%	82.19 ± 0.25 ^K^	1.22 ± 0.10 ^E^	16.16 ± 0.31 ^B^	33.72 ± 1.25 ^G^	28.08 ± 0.61 ^B^
TA15 1%	81.39 ± 0.38 ^J^	1.99 ± 0.08 ^H^	16.10 ± 0.18 ^B^	33.08 ± 0.91 ^G^	28.27 ± 0.44 ^B^
TA30 1%	81.31 ± 0.38 ^J^	2.01 ± 0.11 ^H^	15.64 ± 0.27 ^B^	34.40 ± 1.10 ^G^	27.48 ± 0.57 ^B^
TA 5%	68.27 ± 0.55 ^H^	1.89 ± 0.10 ^G^	25.51 ± 0.15 ^H^	−8.27 ±0.70 ^B^	53.39 ± 0.53 ^G^
TA15 5%	67.27 ± 0.09 ^G^	0.36 ± 0.01 ^B^	20.08 ± 0.01 ^E^	7.03 ± 0.07 ^F^	42.65 ± 0.04 ^C^
TA30 5%	70.60 ± 0.45 ^I^	2.69 ± 0.10 ^I^	25.17 ± 0.07 ^H^	−4.90 ± 0.43 ^C^	50.93 ± 0.31 ^F^
TA 10%	60.32 ± 0.80 ^F^	−0.10 ± 0.03 ^A^	20.07 ± 0.22 ^E^	0.11 ± 0.71 ^D^	47.53 ± 0.56 ^E^
TA15 10%	57.32 ± 0.10 ^D^	6.09 ± 0.01 ^J^	25.78 ± 0.02 ^H^	−20.01 ± 0.07 ^A^	64.25 ± 0.08 ^J^
TA30 10%	60.59 ± 0.15 ^F^	1.01 ± 0.00 ^D^	22.10 ± 0.06 ^G^	−5.72 ± 0.02 ^C^	52.11 ± 0.01 ^G^
TA20%	55.62 ± 0.34 ^C^	−0.13 ± 0.18 ^A^	21.25 ± 0.88 ^F^	−8.13 ± 2.82 ^B^	54.59 ± 2.45 ^H^
TA15 20%	53.72 ± 0.48 ^B^	0.81 ± 0.06 ^C^	19.76 ± 0.12 ^E^	−5.56 ± 0.16 ^C^	52.55 ± 0.18 ^G^
TA30 20%	59.65 ± 0.10 ^E^	−0.16 ± 0.02 ^A^	17.96 ± 0.03 ^C^	5.77 ± 0.07 ^F^	43.01 ± 0.05 ^C^
TA 40%	44.45 ± 0.36 ^A^	1.13 ± 0.01 ^E^	17.67 ± 0.39 ^C^	−8.55 ± 0.84 ^B^	56.78 ± 0.83 ^I^
TA15 40%	60.55 ± 0.05 ^F^	1.70 ± 0.01 ^F^	19.09 ± 0.02 ^D^	3.27 ± 0.08 ^E^	45.05 ± 0.06 ^D^
TA30 40%	55.15 ± 0.32 ^C^	2.10 ± 0.01 ^H^	18.92 ± 0.10 ^D^	−1.61 ± 0.63 ^D^	49.01 ± 0.55 ^E^

Data are expressed as the mean ± SD (*n* = 3). Values with the same capital letter in each column did not show significant differences (*p* < 0.05). Whiteness Index (*WI*) and Yellowness Index (*YI*).

**Table 2 foods-12-03833-t002:** Most important characteristics of the obtained microcapsules.

Samples	MC (%)	EE (%)	IP (h)	PF	*D*[4,3]	PDI
Bulk oil	-	-	2.51 ± 0.15 ^A^	-	-	-
SPI	3.16 ± 0.18 ^A^	60.21 ± 1.06 ^A^	5.35 ± 0.12 ^B^	2.13	10.52 ± 0.46 ^B^	0.89 ± 0.04 ^A^
TA 1%	4.14 ± 0.02 ^B^	75.18 ± 17.84 ^A^	7.56 ± 0.16 ^D^	3.01	1.37 ± 0.00 ^A^	0.93 ± 0.00 ^A^
TA15 1%	3.16 ± 0.05 ^A^	70.38 ± 4.28 ^A^	7.94 ± 0.01 ^D^	3.16	2.54 ± 2.08 ^A^	1.06 ± 0.28 ^A^
TA30 1%	3.24 ± 0.43 ^A^	69.32 ± 1.38 ^A^	7.26 ± 0,10 ^D^	2.89	1.57 ± 0.02 ^A^	0.95 ± 0.00 ^A^
TA 5%	4.59 ± 0.25 ^C^	73.29 ± 8.05 ^A^	11.97 ± 0.63 ^F^	4.77	1.34 ± 0.00 ^A^	0.91 ± 0.01 ^A^
TA15 5%	3.26 ± 0.16 ^A^	65.99 ± 7.67 ^A^	7.62 ± 0.05 ^D^	3.04	1.59 ± 0.00 ^A^	0.95 ± 0.00 ^A^
TA30 5%	5.86 ± 0.13 ^D^	76.95 ± 9.19 ^A^	11.87 ± 0.54 ^F^	4.73	1.39 ± 0.00 ^A^	0.86 ± 0.00 ^A^
TA 10%	3.76 ± 0.44 ^B^	70.07 ± 3.50 ^A^	11.00 ± 0.25 ^E^	4.38	2.18 ± 0.23 ^A^	1.04 ± 0.05 ^A^
TA15 10%	3.82 ± 0.13 ^B^	54.62 ± 3.75 ^A^	6.63 ± 0.06 ^C^	2.64	12.42 ± 0.59 ^B^	1.04 ± 0.01 ^A^
TA30 10%	3.57 ± 0.44 ^B^	67.43 ± 9.12 ^A^	10.90 ± 0.27 ^E^	4.34	13.27 ± 0.72 ^B^	0.97 ± 0.06 ^A^
TA 20%	5.58 ± 0.10 ^D^	61.51 ± 4.97 ^A^	15.10 ± 0.19 ^G^	6.01	21.84 ± 2.45 ^C^	1.18 ± 0.06 ^A^
TA15 20%	3.67 ± 0.05 ^B^	78.38 ± 8.49 ^A^	15.87 ± 0.16 ^H^	6.32	18.10 ± 0.36 ^C^	1.03 ± 0.01 ^A^
TA30 20%	4.06 ± 0.05 ^B^	68.37 ± 0.57 ^A^	15.95 ± 0.25 ^H^	6.35	2.92 ± 1.55 ^A^	1.12 ± 0.20 ^A^
TA 40%	4.74 ± 0.24 ^C^	59.93 ± 0.37 ^A^	21.48 ± 0.28 ^I^	8.55	0.82 ± 0.02 ^A^	0.87 ± 0.03 ^A^
TA15 40%	3.43 ± 0.16 ^A^	76.28 ±13.77 ^A^	27.94 ± 0.88 ^J^	11.12	1.74 ± 0.39 ^A^	0.97 ± 0.04 ^A^
TA30 40%	3.29 ± 0.01 ^A^	63.76 ± 4.01 ^A^	22.17 ± 0.09 ^I^	8.83	1.51 ± 0.06 ^A^	0.95 ± 0.01 ^A^

Moisture content (MC), encapsulation efficiency (EE), induction period (IP), protection factor (PF), De Broucker mean diameter (*D*[4,3]), and polydispersity index (PDI). Data are expressed as the mean ± SD (*n* = 3). Values with the same capital letter in each column did not show significant differences (*p* < 0.05).

**Table 3 foods-12-03833-t003:** Total pore area (TPA) and % porosimetry of samples with 0 and 40% TA.

Samples	TPA [m^2^/g]	% Porosity
SPI	22.10 ^A^	42.15 ^A^
TA 40%	17.75 ^A^	40.50 ^A^
TA15 40%	17.10 ^A^	40.00 ^A^

Data are expressed as the mean ± SD (*n* = 5). Values with the same capital letter in each column did not show significant differences (*p* < 0.05).

**Table 4 foods-12-03833-t004:** Fatty acid composition profile, moisture content (MC), water activity (aw), De Broucker mean diameter (*D*[4,3]) and the polydispersity index (PDI) of samples (0 day) and stored for 180 days.

Samples	Days	Saturated	C18:1	C18:2	C18:3	MOC (%)	aw	*D*[4,3]	PDI
SPI	0	96.04 ± 2.19 ^Aa^	64.18 ± 0.42 ^Aa^	191.47 ± 0.96 ^Aa^	649.20 ± 3.03 ^Ba^	2.16 ± 0.02 ^Aa^	0.168 ± 0.001 ^Aa^	12.27 ± 0.04 ^Ad^	1.10 ± 0.00 ^Aa^
	180	230.16 ± 2.39 ^B**c**^	168.34 ± 2.21 ^B**b**^	228.65 ± 2.08 ^B**c**^	333.22 ± 1.54 ^A**a**^	3.29 ± 0.13 ^B**a**^	0.285 ± 0.008 ^B**a**^	217.27 ± 42.86 ^B**b**^	2.42 ± 0.09 ^B**a**^
TA 5%	0	94.38 ± 0.27 ^Aa^	64.72 ± 0.42 ^Aa^	194.50 ± 0.43 ^Ba^	645.49 ± 3.02 ^Ba^	3.45 ± 0.35 ^Ab^	0.170 ± 0.001 ^Aa^	2.53 ± 0.08 ^Ac^	4.38 ± 0.17 ^Bc^
	180	97.95 ± 0.39 ^B**b**^	65.42 ± 0.53 ^A**a**^	192.55 ± 0.40 ^A**b**^	636.14 ± 0.40 ^A**b**^	3.45 ± 0.35 ^A**a**^	0.206 ± 0.008 ^A**b**^	15.55 ± 0.25 ^B**a**^	1.21 ± 0.08 ^A**a**^
TA15 5%	0	98.61 ± 0.08 ^Ba^	65.50 ± 0.65 ^Ba^	193.70 ± 0.43 ^Aa^	628.38 ± 3.65 ^Aa^	3.27 ± 0.47 ^Ab^	0.178 ± 0.002 ^Aa^	2.61 ± 0.03 ^Ac^	5.32 ± 0.37 ^Bd^
	180	93.96 ± 1.08 ^A**a**^	63.22 ± 0.27 ^A**a**^	194.04 ± 0.67 ^Ab^	646.38 ± 4.88 ^B**c**^	3.67 ± 0.40 ^A**a**^	0.227 ± 0.008 ^B**c**^	18.81 ± 1.30 ^B**a**^	0.95 ± 0.06 ^A**a**^
TA 40%	0	98.49 ± 0.08 ^Ba^	66.42 ± 0.65 ^Aa^	194.39 ± 0.44 ^Ba^	642.11 ± 3.02 ^Aa^	2.28 ± 0.13 ^Aa^	0.157 ± 0.002 ^Aa^	2.24 ± 0.02 ^Aa^	3.58 ± 0.27 ^Ab^
	180	90.44 ± 2.35 ^A**a**^	61.07 ± 1.92 ^A**a**^	186.96 ± 1.05 ^A**a**^	655.79 ± 3.44 ^B**d**^	3.28 ± 0.14 ^B**a**^	0.163 ± 0.003 ^B**d**^	22.07 ± 3.82 ^B**a**^	1.02 ± 0.08 ^A**a**^
TA15 40%	0	94.00 ± 0.27 ^Ba^	63.90 ± 0.42 ^Ba^	192.84 ± 0.97 ^Ba^	647.41 ± 3.03 ^Aa^	2.83 ± 0.16 ^Ab^	0.175 ± 0.002 ^Ba^	2.37 ± 0.02 ^Ab^	6.03 ± 0.06 ^Be^
	180	92.16 ± 1.02 ^A**a**^	61.30 ± 1.87 ^A**a**^	187.47 ± 1.04 ^A**a**^	655.49 ± 3.42 ^B**d**^	3.39 ± 0.18 ^B**a**^	0.168 ± 0.001 ^A**d**^	19.33 ± 0.33 ^B**a**^	0.95 ± 0.03 ^A**a**^

Data are expressed as the mean ± SD (*n* = 3). Values with the same capital letter in each column did not show significant differences (*p* < 0.05) between 0 and 180 days for each sample. Values with different lowercase letters show significant differences (*p* < 0.05) between samples at day 0 and bold lowercase letters between samples at day 180.

**Table 5 foods-12-03833-t005:** Fatty acid composition of samples after simulated gastrointestinal digestion.

Sample	% OA	Saturated (mg/g Oil)	C18:1 (mg/g Oil)	C18:2 (mg/g Oil)	C18:3 (mg/g Oil)
Bulk oil	100.00 ± 0.06 ^D^	101.99 ± 25.01 ^A^	74.62 ± 6.53 ^A^	209.51 ± 26.04 ^A^	613.89 ± 58.08 ^A^
SPI	95.17 ± 0.46 ^C^	99.49 ± 4.45 ^A^	66.43 ± 3.37 ^A^	202.80 ± 1.71 ^A^	612.44 ± 31.11 ^A^
TA 40%	48.45 ± 0.30 ^B^	105.87 ± 8.22 ^A^	81.39 ± 0.42 ^A^	192.69 ± 18.31 ^A^	638.51 ± 39.52 ^A^
TA15 40%	35.94 ± 2.73 ^A^	89.04 ± 23.35 ^A^	71.64 ± 4.36 ^A^	204.48 ± 4.44 ^A^	632.17 ± 22.17 ^A^

Data are expressed as the mean ± SD (*n* = 3). Values with the same capital letter in each column did not show significant differences (*p* < 0.05).

**Table 6 foods-12-03833-t006:** Total polyphenol content determined by Folin–Ciocalteu method and antioxidant capacity determined by FRAP, TEAC, and DPPH methods for undigested and digested samples.

Sample	TPC (mg GAE/g)	FRAP (mg TE/g)	TEAC (mg TE/g)	DPPH (mg TE/g)
Undigested				
SPI	23.40 ± 9.40 ^b^*	<LOD	<LOD	<LOD
TA 40%	238.10 ± 13.83 ^a^*	554.22 ± 20.91 ^a^	594.89 ± 16.61 ^a^*	528.88 ± 33.40 ^ab^*
TA15 40%	227.07 ± 19.23 ^a^*	569.74 ± 30.85 ^a^*	625.75 ± 34.56 ^ab^*	469.63 ± 86.11 ^b^*
Digested				
SPI	46.62 ± 6.59 ^C^*	<LOD	<LOD	<LOD
TA 40%	322.09 ± 10.08 ^B^*	663.61 ± 132.02 ^AB^	917.25 ± 61.69 ^AB^*	818.92 ± 64.94 ^AB^*
TA15 40%	382.35 ± 95.43 ^A^*	805.82 ± 118.75 ^A^*	1035.17 ± 232.23 ^A^*	916.01 ± 180.35 ^A^*

Data are expressed as the mean ± SD (*n* = 3). Different lowercase letters indicate significant differences (*p* < 0.05) between undigested fractions for the same method. Different capital letters indicate significant differences (*p* < 0.05) between digested fractions for the same method. * Indicates paired differences between undigested and digested fractions for the same sample. < LOD means “under limit of detection”.

## Data Availability

Data is contained within the article.

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
