# Peer review of "Chia Oil Microencapsulation Using Tannic Acid and Soy Protein Isolate as Wall Materials"

_foods, 2023, doi:10.3390/foods12203833_

Round 1
Reviewer 1 Report
Comments and Suggestions for Authors
Dear Authors,
your paper has been sent for my consideration. Unfortunately, due to numerous imperfections, it cannot be processed further.
First, it is not clear what and how the authors did. There is no key information about sample preparation (once? many times? is it repeatable?). There is no information what they analyzed for biological activity (oil? capsules? how were the active substances extracted?)
The collection and use of human biological material (saliva) requires the consent of the ethics committee (no data).
The introduction needs to be refined (comments in the PDF file).
There is basically no discussion, just citing the literature. Why were the observed phenomena not precisely explained? And the reasons for these changes?
I have included other comments in a PDF file.
I'm sorry, but the manuscript requires many changes and cannot be published in a journal with such a high IF in its current form.

Author Response
We thank to reviewer 1 for taking the time to review our manuscript and provide valuable feedback. These suggestions and corrections will help to improve the quality of the manuscript. We appreciate your input and will address each of your comments in a point-to-point reply letter in a attached file.

Reviewer 2 Report
Comments and Suggestions for Authors
To whom it may concern,
The current research article entitled "Chia oil microencapsulation using tannic acid and soy protein 2 isolate as wall materials" is about the concept of encapsulating bioactive compound to prevent from damage during the storage. This is a well-written manuscript and can be considered for publication after minor corrections. My specific comments are attached as a pdf file. Thank you.

Author Response
Thank to Reviewer 2 for taking the time to review our manuscript and provide valuable feedback that helps to improve the quality of the manuscript. We appreciate your input and will address each of your comments in a poin-to-point reply letter in an attached file.

Round 2
Reviewer 1 Report
Comments and Suggestions for Authors
Dear Authors, I have read the replies to my comments. My doubts were explained in great detail and key missing information was added. I realize that each manuscript has its strengths and weaknesses. However, this one has been corrected in a way. There are still text editing issues (typos), but this can be corrected during production.
Reviewer 2 Report
Comments and Suggestions for Authors
To whom it may concern,
The manuscript entitled "Chia oil microencapsulation using tannic acid and soy protein isolate as wall materials" has been revised well. However, there are still some minor gramatical errors should be addressed.